# Individualised screening for diabetic retinopathy: the ISDR study—rationale, design and methodology for a randomised controlled trial comparing annual and individualised risk-based variable-interval screening

Deborah M Broadbent,[1,2] Christopher J Sampson,[3] Amu Wang,[1] Lola Howard,[4] Abigail E Williams,[4] Susan U Howlin,[4] Duncan Appelbe,[4] Tracy Moitt,[4] Christopher P Cheyne,[4,5] Mehrdad Mobayen Rahni,[1] John Kelly,[6] John Collins,[6] Marta García-Fiñana,[4,5] Irene M Stratton,[7] Marilyn James,[3] Simon P Harding,[1,2] for the ISDR Study Group

**Correspondence to**
Dr Deborah M Broadbent; deborah.broadbent@rlbuht. nhs.uk

## ABSTRACT

**Introduction** Currently, all people with diabetes (PWD) aged 12 years and over in the UK are invited for screening for diabetic retinopathy (DR) annually. Resources are not increasing despite a 5% increase in the numbers of PWD nationwide each year. We describe the rationale, design and methodology for a randomised controlled trial (RCT) evaluating the safety, acceptability and cost-effectiveness of personalised variable-interval risk-based screening for DR. This is the first randomised trial of personalised screening for DR and the largest ophthalmic RCT in the UK.

**Methods and analysis** PWD attending seven screening clinics in the Liverpool Diabetic Eye Screening Programme were recruited into a single site RCT with a 1:1 allocation to individualised risk-based variable-interval or annual screening intervals. A risk calculation engine developed for the trial estimates the probability that an individual will develop referable disease (screen positive DR) within the next 6, 12 or 24 months using demographic, retinopathy and systemic risk factor data from primary care and screening programme records. Dynamic, secure, real-time data connections have been developed. The primary outcome is attendance for follow-up screening. We will test for equivalence in attendance rates between the two arms. Secondary outcomes are rates and severity of DR, visual outcomes, cost-effectiveness and health-related quality of life. The required sample size was 4460 PWD. Recruitment is complete, and the trial is in follow-up.

**Ethics and dissemination** Ethical approval was obtained from National Research Ethics Service Committee North West – Preston, reference 14/NW/0034. Results will be presented at international meetings and published in peer-reviewed journals. This pragmatic RCT will inform screening policy in the UK and elsewhere.

**Trial registration number** ISRCTN87561257; Pre-results.

### Strengths and limitations of this study

► Our study addresses one of the primary current issues in the field of screening for diabetic retinopathy: the safety of extending the interval between episodes.
► Our study has significant novelty applying a personalised approach to a whole population intervention, using routinely collected data from primary and secondary care, a risk calculation engine that allows generalisation and evaluated in the first RCT conducted on screening for diabetic retinopathy.
► Strengths include the substantial involvement of people with diabetes in the design and implementation, large numbers of participants for tight confidence limits, pragmatic solutions for variable quality routine National Health Service data and the independent analysis by an accredited clinical trials unit.
► The risk calculation engine has been internally validated but will require further external validation before implementation could be considered elsewhere.
► The trial only recruits people currently attending screening and does not identify the impact for people who currently do not attend.

## INTRODUCTION
### Background

Diabetes mellitus (DM) is a life-long condition associated with the development of various macrovascular and microvascular complications, including diabetic retinopathy (DR) and maculopathy (a subgroup within DR). These are progressive conditions of the retina and macula, which can lead to visual impairment (VI) and blindness. There are an

estimated 3.8 million people with diabetes (PWD) aged 16 years and over in England,[1] of whom almost 1 million are undiagnosed. Nearly all people with type 1 diabetes[2] and over 60% with type 2 diabetes[3] will develop some degree of DR after 20 years of having diabetes.

DR affects people of all ages and is the most common cause of blindness in people of working age in most developed countries worldwide. In the UK, this position has changed. Liew et al[4] reviewed the causes of blindness certifications in working age adults in England and Wales in 2009–2010 and reported that DR had dropped to the second most common cause compared with 10 years earlier. Tighter control of glycaemia and hypertension will have contributed to this, but it is probable that screening for DR has also played a key role.

The early stages of DR (background and mild pre-proliferative) are asymptomatic and do not require any treatment. Sight-threatening diabetic retinopathy (STDR; an umbrella term that encompasses sight-threatening levels of retinopathy and maculopathy) requires close monitoring by an ophthalmologist, and sometimes treatment, to prevent VI.

Treatment (by laser photocoagulation or intravitreal injections) aims to stop progression and stabilise retinopathy. Treatment cannot always reverse the process once vision is lost. Screening is therefore recommended to identify STDR at the optimal time-point for treatment. Previous studies have shown that screening for DR is a highly cost-effective intervention[5 6]: the higher the take-up rates for screening, the higher the cost-effectiveness.[7]

Annual systematic screening for DR for all PWD aged 12 years or older was introduced across the UK by 2007. A 1-year interval between screening invitations (screen interval) was based on expert opinion, rather than direct evidence. In 2016, the UK National Screening Committee recommended that PWD at low risk of sight loss could be screened every 2 years, but this has not yet been adopted. This stratified approach was based on evidence from a large observational study in one English programme and a cost-effectiveness analysis.[8] The findings were validated in the Welsh, Scottish and Northern Ireland programmes and four more English programmes. Stratified screening is due to be implemented in Scotland.

Available evidence shows that many people at low risk of developing referable DR between annual screening appointments could safely be screened less often,[9] while others are at high risk and might benefit from more frequent screening.[10] In 1996, Davies et al[11] used simulation modelling of published data and concluded that biennial screening could be considered where patient compliance and screening sensitivities were both high. Data on 10-year incidence from the Liverpool Diabetic Eye Study—for a population of people with type 2 DM and enrolled in a systematic screening programme—suggested that a 3-yearly screening interval could be adopted for PWD with no retinopathy at baseline, but yearly or more frequent screening was recommended for people with higher grades of retinopathy or insulin use.[12]

Similar results were shown for type 1 DM.[13] In Sweden, biennial screen intervals have been used for some time for subjects without retinopathy.[14] A study carried out in Malmo prospectively followed people with type 2 DM and no retinopathy and concluded that it appeared safe to adopt 3-year intervals[15] as suggested by the Liverpool group. However, this group of PWD were compliant (only 9% did not attend for follow-up) and had a short duration of diabetes (6±6 years) and good control (glycated haemoglobin (HbA1c) 6.4%±1.5% at baseline and 6.3%±1.3% at 3-year follow-up).

Two studies have looked at stratified screening and shown that the risk of progression to STDR is significantly higher for those with background DR (BDR) in both eyes than those with BDR in only one or in neither eye.[16 17] The first of these two studies suggested that combining the results from two consecutive years of photographic screening enabled estimation of the risk of future development of STDR. People with no DR on two consecutive visits were deemed to be low risk.

So why hasn't an extended interval in screening for STDR been more widely adopted? There has been considerable concern about the safety of extending screen intervals and, to date, no randomised controlled trial (RCT) has reported.[9] The experience of Sweden and Scotland is reassuring, but the populations are much smaller and generally better engaged. If people receive the impression that they are at low risk, they may disengage with other aspects of diabetes care. Despite encouraging findings from our group in 2003, in the absence of the required safety data, it appears unwise to recommend 2-year or 3-year intervals for PWD at low risk without further evidence.

## Rationale for an RCT

With this in mind, we designed an RCT to investigate the safety of extending screening intervals in low-risk PWD. We included the emerging theories and technologies of risk prediction and personalisation to develop a risk-based variable-interval screening approach and incorporated an economic evaluation.

In addition to severity of retinopathy, the risk of development and progression of DR to a level that requires treatment is related to age, gender,[1 2] duration of diabetes, glycated haemoglobin (HbA1c) levels,[18–23] blood pressure,[24–26] lipid levels[27] and proteinuria.[28] If the contribution of each risk factor to overall risk (as well as the overall contribution in combination) could be calculated, and data on each risk factor were available for individuals, individual risks could be estimated and screening frequencies could be tailored to the level of personal risk.[29]

We developed a risk calculation engine (RCE) using a longitudinal dataset from our local diabetic population to predict the risk of developing STDR and have reported confident estimates of the risk of having STDR.[30] In Iceland, a similar risk algorithm has also been developed to estimate the risk of development of STDR but

based on modelling of historic epidemiological data on a limited number of risk factors for DR.[31] In the Netherlands, a screening model based on patients' risk has been validated in 3319 people with type 2 diabetes as part of the Hoorn Study.[32] We embedded a patient and public involvement (PPI) group in the development of the RCE and subsequent design of the trial. There has been scant research into people's understanding of screening and their views about introducing variable screening intervals. Yeo et al[33] reported that extended intervals may be acceptable to the majority of PWD if there was adequate evidence to support such a change.[33] We found that over a series of workshops the PPI volunteers became expert in the field and made important design decisions.

We believe that screening for referable DR at intervals based on individual risk could be both safe and cost-effective compared with annual screening. In this paper, we report the key features of the design of an RCT designed to test this hypothesis.

## Funding and ethics approval

The Individualised Screening for Diabetic Retinopathy (ISDR) RCT is part of a larger programme of applied research funded by a £2.1 million grant from the National Institute for Health Research (NIHR) Programme Grants for Applied Research programme – reference RG-PG-1210–12016. SH is the chief investigator for the seven workstream programmes, and DMB is the principal investigator for the RCT. The RCT is supported by the Clinical Trials Research Centre in Liverpool, which provides information systems support, developed electronic case report files and manage the data. There are three trial oversight committees: a trial management group, a trial steering committee and the independent data and safety monitoring committee. The composition and membership of the committees was approved by NIHR. The study protocol,[34] patient information sheets and consent forms have received ethical approval from the National Research Ethics Service Committee North West – Preston (reference: 14/NW/0034). Recruitment to the trial was supported by research staff from Clinical Research Network North West Coast, Liverpool Diabetic Eye Screening Programme (LDESP) staff and trained student volunteers.

## Trial status

The trial opened to recruitment on the 12 November 2014. The first patient was randomised on the 19 November 2014. The trial was closed to enrolment on the 31 May 2016. The target of 4460 patients was reached. The programme grant completes on 30 April 2019.

## Patient and public involvement

We have included members of our PPI Group in all stages of the design of the trial. Our PPI group consists of seven individuals and has met at regular intervals throughout the lifetime of the trial. Individuals are active participants in the programme and trial steering committees,

programme investigators committee and trial management committee. Individuals were involved in the concept and development of the research questions, design of the trial and the grant application. One patient is a coinvestigator and was closely involved in designing the intervention including reviewing the potential burden. The group was involved in the choice of risk factors for consideration for the RCE and secondary outcome measures. They developed a set of patient-centred outcomes, most notably adding VI and need for treatment, which they viewed as particularly important to patients. They considered the chance of missing disease if screen intervals were extended and after several structured sessions settled on a 2.5% risk as an acceptable limit. They were not involved in actual recruitment to the trial, but they visited the screening centres and observed recruitment and gave constructive comments, which improved the process.

Regarding dissemination of results, these will be sent to the participants' general practitioners. Participants have also been advised to let the researcher know if they would like a summary of the results themselves.

## Participants, interventions and outcomes

### Trial design

The ISDR RCT had 1:1 allocation to an indivdualised risk-based screening recall (6, 12 or 24 months) or annual screening (current routine care).

### Objectives

The objective of the ISDR RCT is to evaluate the safety, acceptability and cost-effectiveness of personalised screening in a whole population setting. The main aim is to assess the relative safety of variable-interval screening as measured by attendance rate. We will test the hypothesis that the difference in attendance rates between the two pathways is within an acceptable equivalence margin.

### Study setting

This single site trial is being conducted in all seven screening clinics in the LDESP, which is part of the English National Diabetic Eye Screening Programme (NDESP).

### Study population

PWD registered with a GP whose postcode is in the city boundaries of Liverpool and attending for screening for DR were invited to enter the trial. The inclusion and exclusion criteria are given in table 1.

The schematic of the ISDR RCT study design is given in figure 1.

### Data flows

Data flows are shown in figure 2. Data on the participant's retinopathy status feeds automatically from the screening software (OptoMize, EMIS health) into the data warehouse (DW) and then to the randomisation and RCE programmes on a daily basis. The ISDR DW automatically populates the majority of the fields in the baseline and follow-up electronic CRFs (OpenClinica), including data that allowed randomisation to occur.

**Table 1** Inclusion and exclusion criteria in the personalised variable-interval risk-based screening for diabetic retinopathy randomised controlled trial

| Inclusion criteria | Exclusion criteria |
|---|---|
| Adults, young people and children who are aged ≥12 years. | Under age 12 years. |
| Due to be offered an appointment for retinal screening during the recruitment period. | |
| Registered with a participating GP practice. | Are not registered with a participating GP practice. |
| Are included in the study data warehouse (have not opted out). | Have opted out from the study data warehouse. |
| Have no retinopathy or have retinopathy and maculopathy less than the definition of screen positive diabetic retinopathy. | Have screen positive diabetic eye disease or significant other eye disease requiring referral to the HES. |
| Have gradeable digital retinal images in both eyes. | Are ineligible for screening for whatever reason, including having ungradable digital retinal images, which includes patients who have only one eye or an ungradable eye with no visual potential. |
| Give their informed consent for participation. | Do not give consent for participation in the RCT. |
| Are not involved in any trial investigating a treatment aiming at preventing or modifying the development of STDR. | Are involved in any trial investigating a treatment aiming at preventing or modifying the development of STDR. |

GP, general practitioner; HES, hospital eye service; STDR, sight-threatening diabetic retinopathy.

## Primary endpoint

The primary outcome is the attendance rate for follow-up screening in the two arms of the study. Non-attendance is defined as failure to attend any screening appointments within 90 days of the expected follow-up date, irrespective of how many appointments they had been sent.

## Secondary endpoints

► Number of cases of STDR detected.
► Retinopathy level at screening.
► Maculopathy level at screening.
► Number of false positive screening episodes.
► Number of screening appointments.
► Number of diabetes assessment clinic appointments.
► Number of other eye appointments for DR.
► Visual acuity (log of the minimum angle of resolution [logMAR]).
► New VI (≥+0.30 and ≥+0.50 logMAR),
► New VI (≥+0.30 and ≥+0.50 logMAR) which, in the opinion of an experienced clinician, is due to DR.
► Number of missed appointments to screening.
► Patient acceptability measures (using a questionnaire designed for the RCT).
► Quality-adjusted life years (QALYs) estimated using EQ-5D-5L and Health Utilities Index Mark 3 (HUI3).
► Cost per QALY.

## Intervention, assessments and procedures

The ISDR DW is a relational database that stores both the data and how the data are related. The Liverpool GPs gave permission for their PWD to be directly contacted to give permission for the data held in primary care relating to their diabetes to be transferred to the DW. The DW was established to collate and dynamically link individual risk factor data from GP practices, STDR outcome data from the DR assessment clinic at St Pauls Eye Unit and the LDESP screening software (OptoMize). This central data repository was used to develop the locally applicable RCE described above using a continuous Markov model.

The RCE was developed using routinely collected primary and secondary care data from 11 806 PWD (46 525 screening episodes) in the LDESP. A detailed description of the RCE has already been published.[30] In brief, the final covariates for the RCE were selected using a three-step decision process. A number of risk factors for developing STDR were identified in the published literature and in preliminary work in Liverpool. These potential covariates were reviewed by the clinical team and the PPI group, and potential additional covariates were suggested. Potential covariates with a missingness rate of less than 20% in the DW were identified. Then a statistical evaluation of the predictive value of each remaining covariate was undertaken. The covariates selected were disease state (current retinopathy levels in both eyes); age; duration of diabetes; glycated haemoglobin; systolic blood pressure; and total cholesterol. The results suggested that implementing personalised risk-based intervals would reduce the number of screen episodes by 30%.

Our PPI group defined the degree of risk (up to 2.5% risk of developing screen positive DR) acceptable to PWD, allowing assignment of individuals to screen intervals at the time points 6, 12 or 24 months. The choice of these intervals was based on a review of the available literature, our data and consensus between the PPI and research teams. Following each negative screening outcome, individuals are assigned to the longest recall period up to 24 months at which their risk estimation would not exceed the 2.5% threshold.

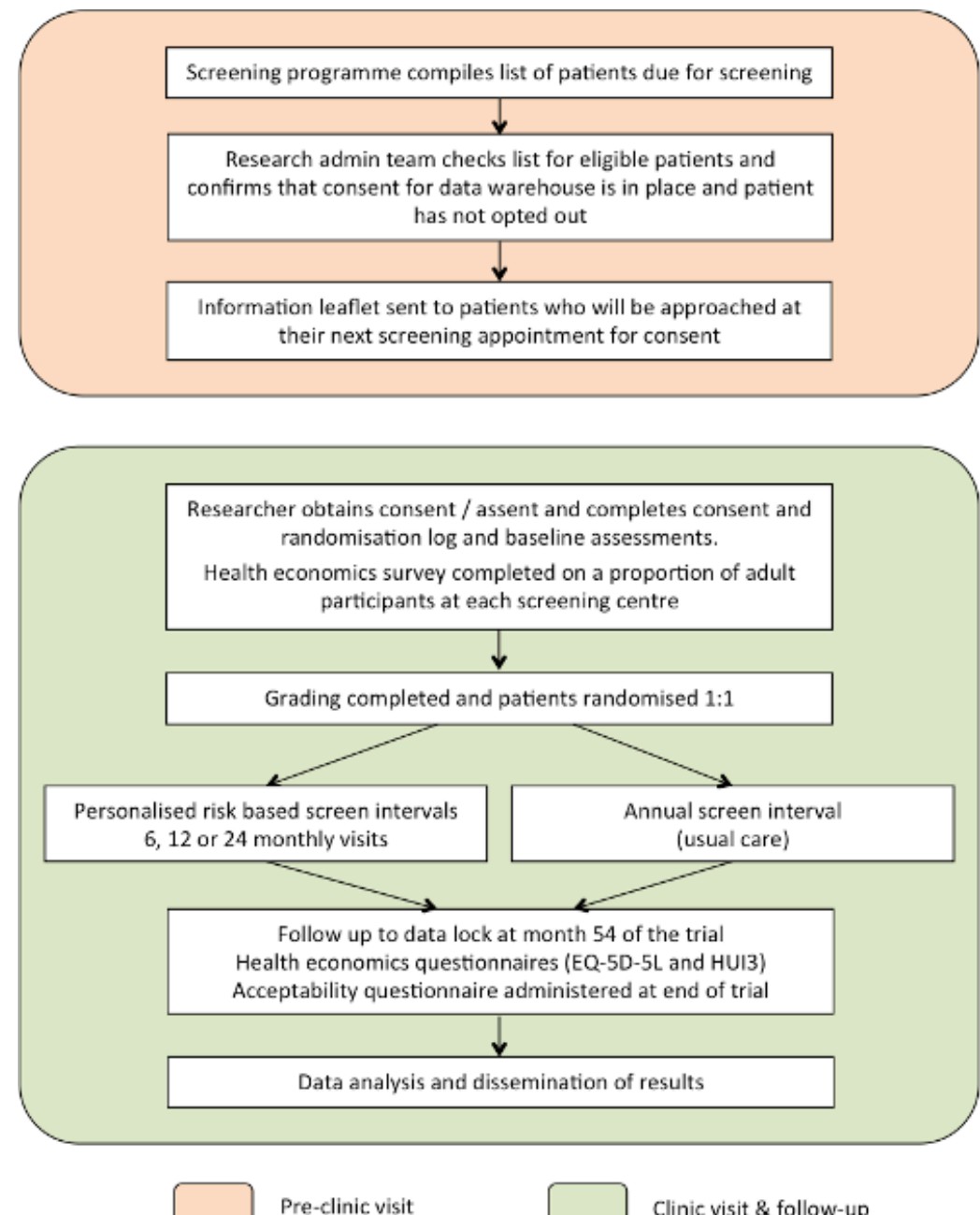

**Figure 1** Schematic of the ISDR RCT trial design.

## Consent

Individuals received a patient information leaflet with all the details of the trial with their screening appointment reminder letter. On attending the screening clinic, they were approached by trained researchers to establish whether they wished to participate in the trial. If they expressed an interest, the researcher took written informed consent and enrolled them into the trial. If an individual subsequently requested to be withdrawn from the trial, they reverted to routine care. Reasons for, and level of, withdrawal were collected.

## Randomisation

Randomisation could only be completed once the individual's retinal images had been graded, as it was only possible at this point to complete the inclusion and exclusion criteria. PWD who screened positive due to DR, other eye disease or ungradable images were therefore consented but were not included in the trial.

Eligible PWD were randomly allocated to the personalised risk-based screening recall or the usual fixed annual interval. Participants were stratified by clinic and age (<16 and ≥16 years old). The screening clinics in Liverpool had different proportions of PWD in hard to reach groups, such as minority ethnic groups, attendance rates and social deprivation indices.

After randomisation the RCE was used to generate the individual's recall period in the personalised arm of the trial and the data was sent back to the screening software.

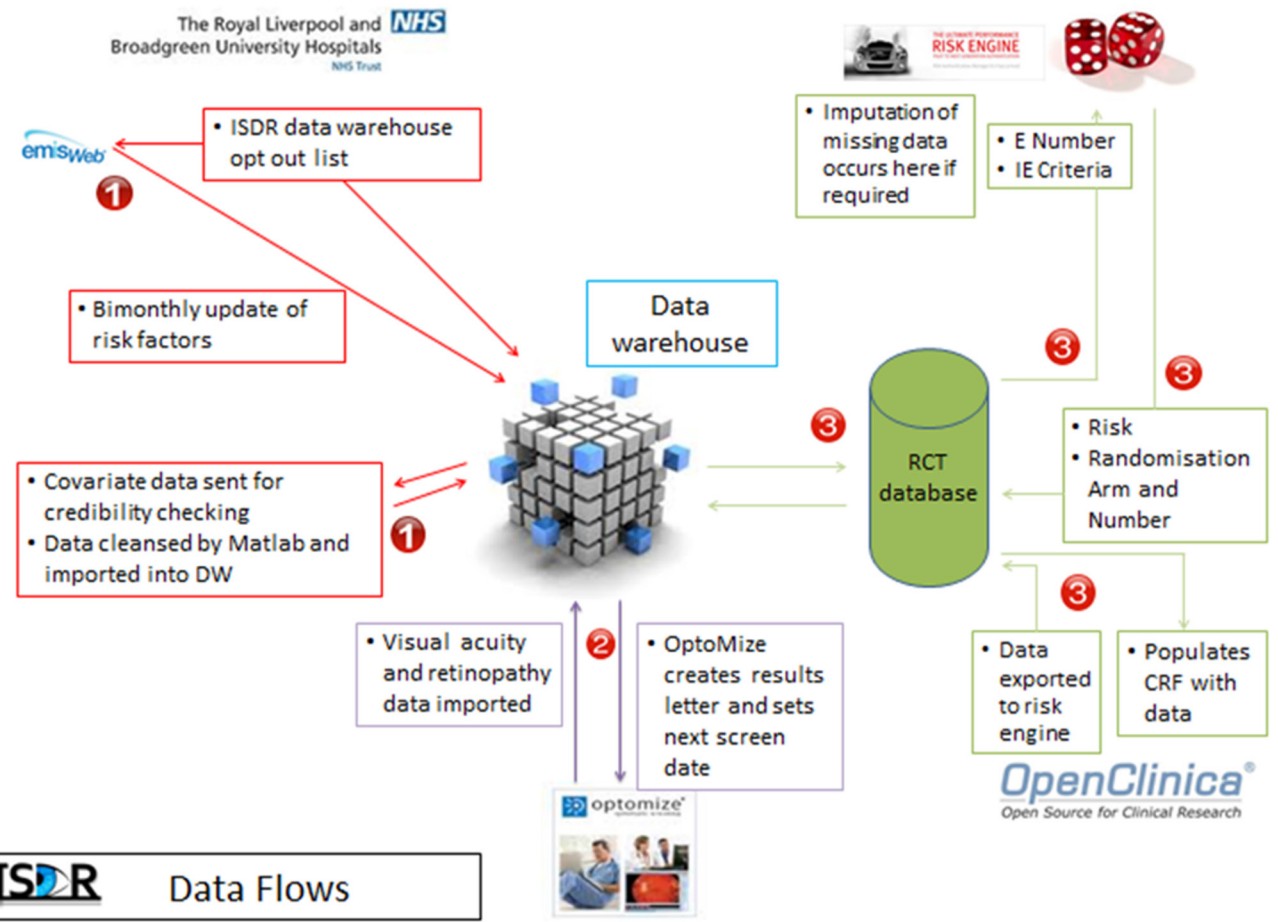

**Figure 2** Data flows. In step 1, data for consented participants are requested from OptoMize, passed to the DW and cleaned prior to storage. In step 2, data are exchanged between the DW and OptoMize (subjects whose risk needs to be calculated are sent to the DW, subjects for whom their risk and therefore recall interval has been calculated is returned to OptoMize for appointment letter generation). In step 3, the participants' risk is calculated (when all the covariates are available); if randomisation is required, they are then randomised. The data are then stored in the study database and the DW. All processes in steps 1 and 2 involve identifiable data; the processes in step 3 all use pseudanonomised data for security reasons (the trial team with access to the trial database do not have a need to see raw identifiers). Step 1 occurs on a bimonthly period, steps 2 and 3 occur on a daily basis. Under ideal conditions, it takes 3 days for the data to pass through all parts of steps 1 and 2; this is due to an air gap (manual transfer) at step 2 between the DW and the National Health Service systems. CRF, case report form; DW, data warehouse.

### Study visits and assessments

At each follow-up visit, PWD randomised to the personalised risk-based screening arm have their risk recalculated and the next recall interval determined accordingly.

### Health economics

Table 2 shows the trial intervention diagram.

A health economics questionnaire, incorporating the health-related quality of life (QoL) questionnaires EQ-5D-5L (an instrument to meaure health state developed by the EuroQol group)[35] and Health Utility Index 3 (HUI3)[36] and a bespoke visit questionnaire,[37] was completed by the first 868 eligible PWD enrolled into the study (a minimum of 700 PWD was required to provide sufficient data). The QoL questionnaires are repeated at every subsequent follow-up visit for those PWD who completed it at baseline.

### Sample size and planned analyses

The primary aim is to assess the relative safety of personalised risk-based interval screening as measured by equivalence in attendance rates. The analysis will test the hypothesis that the difference in attendance rates at the first follow-up between the two pathways is within the acceptable range $\delta$=0.05. If the bounds of the 90% CI for the difference in attendance rate are found to be within the confidence limits ($-\delta, \delta$ (ie, (−0.05 to 0.05)), the results would support equivalence.

The estimated minimum number of patients required is 3940. With an expected loss to follow-up rate of 6% per year due to death and other exclusion from screening (note that non-attendance is the primary outcome and therefore is not factored here), the target for recruitment (randomised into the trial) was 4460 patients (4460*0.94*0.94=3940).

**Table 2** Trial intervention diagram

| Procedures | Screening | Baseline | Randomisation | Follow-up schedule | | | | | | Study completion | Premature discontinuation |
|---|---|---|---|---|---|---|---|---|---|---|---|
| | | | | 6months* | 12months* | 18months* | 24months* | 30months* | 36months* | | |
| Signed consent form | X | | | | | | | | | | |
| Visual acuity (logMar) | | X | | X | X | X | X | X | X | | |
| Visit questionnaire | | X | | | | | | | | | |
| QoL questionnaire† (HUI3 and EQ5D) | | X | | X | X | X | X | X | X | | |
| Assessment of eligibility criteria | X | X | | | | | | | | | |
| Randomisation | | | X | | | | | | | | |
| Digital imaging | | X | | X | X | X | X | X | | | |
| Acceptability questionnaire | | | | | | | | | X | | |

*PWD are randomised to two arms. In one arm, individuals attend at 12-month intervals. In the other arm, individuals attend at 6, 12 or 24 monthly intervals.
†The EQ-5D-5L and HUI3 data will be collected on around 700 eligible PWD and repeated at each visit for those individual PWD. Data will be stratified to balance equal numbers of PWD in each arm and representation from all screening sites.
HUI3, Health Utilities Index Mark 3; QoL, quality of life; PWD, people with diabetes.

A secondary aim is to investigate whether personalised screening can be considered as non-inferior in detection of STDR when compared with annual screening. The STDR detection rate predicted for the usual care pathway during the 2-year follow-up is approximately 6%, based on data from the LDESP. The sample size required to address the first question (n=4460 patients randomised with 3940 patients retained after 2 years from baseline) will permit us to test for non-inferiority in STDR detection with a maximum allowable reduction of 1.5% of the personalised care pathway compared with standard care, with 5% significance level and power between 60% and 65%.

We will undertake subgroup analyses to assess differences in attendance rates between the two arms for the three different retinopathy groups (the risk groups will be defined based on the individual baseline estimated risk of developing STDR). A logistic mixed-effects model that takes into account the patient's covariate information over time (including retinopathy level, HbA1c, systolic blood pressure, total cholesterol, disease duration, and age), and screening clinic (clustered data) will be fitted with attendance (yes/no) at the first screening visit as the outcome variable. The random effects of the model will account for the variability by screening clinic.

Using screening activity data stored in the ISDR DW and information collected from the visit questionnaire and EQ-5D-5L and HUI3 responses, we will estimate the cost per QALY within the study period associated with risk-based and annual screening from an NHS perspective and where possible incorporate a broader perspective. We will present bootstrapped incremental cost-effectiveness ratios and cost-effectiveness acceptability curves to characterise the uncertainty associated with our estimates.

We will apply sensitivity analyses to check the sensitivity of the results on the assumption that missing data are missing at random. Different scenarios for missing data mechanisms will be explored.

## CONCLUSIONS

This paper describes the design of an RCT to evaluate the feasibility, safety, acceptability and cost-effectiveness of personalised variable-interval risk-based screening compared with fixed annual interval screening. Safety will be measured by the effect on attendance rates to screening, rates and severity of DR, visual outcomes and impact on general diabetes care. As far as we are aware this is the only study to investigate the impact of implementing personalised screening for DR in a randomised controlled trial.

### Trial sponsors

University of Liverpool Research Support Office, 1st floor Duncan Building, Daulby Street, Liverpool L69 3GA. Tel: 0151 706 3702.

Research and Innovation Department, Royal Liverpool and Broadgreen University Hospitals Trust, Prescot Street, Liverpool L7 8XP 0151 706 2000.

University of Liverpool: UoL000994 Research and Innovation Department Royal Liverpool and Broadgreen University Hospitals Trust Prescot Street, Liverpool L7 8XP 0151 706 2000, Trust RD&I: 4660.

**Author affiliations**
[1]Department of Eye and Vision Science, Institute of Ageing and Chronic Disease, University of Liverpool, Liverpool, UK
[2]St Pauls Eye Unit, Royal Liverpool University Hospital, Liverpool, UK
[3]Division of Rehabilitation and Ageing, School of Medicine, University of Nottingham, Nottingham, UK
[4]Department of Biostatistics, Clinical Trials Research Centre, University of Liverpool, Liverpool, UK
[5]Department of Biostatistics, University of Liverpool, Liverpool, UK
[6]Patient and Public Involvement Group, Liverpool, UK
[7]Gloucestershire Retinal Research Group, Cheltenham General Hospital, Cheltenham, UK

**Acknowledgements** We would like to acknowledge the outstanding attention to detail from the Individualised Screening for Diabetic Retinopathy (ISDR) administrative team and the Clinical Trials Research Centre (CTRC). We would like to thank the North West Coast CRN and our other trained researchers for their invaluable help in recruiting to target. We would also like to thank the Liverpool Diabetic Eye Screening Programme for their support and assistance. We would particularly like to thank our patient and public involvement (PPI) members for their invaluable advice and enthusiastic support for the study. CJS is currently affiliated to the Office of Health Economics, Southside, 7th Floor, 105 Victoria Street, London SW1E 6QT.

**Collaborators** ISDR Study GroupSimon P Harding (Study Group Chair), Deborah M Broadbent (Trial PI), Anthony C Fisher, Mark Gabbay, Marta García-Fiñana, Marilyn James, Tracy Moitt, John R Roberts, Daniel Seddon, Irene M Stratton, Paula Williamson, Duncan Appelbe, Andy Ovens, Lola Howard, Susan U Howlin, Kate Silvera, Ayesh Alshukri, Abigail E Williams, Christopher P Cheyne, Antonio Eleuteri, Christopher Grierson, Mehrdad Mobayen-Rahni, Christopher J Sampson, David Szmyt, Clare Thetford, Pilar Vazquez Arango, Amu Wang, Helen Cooper, John Collins, John Kelly, Nathalie Massat, Gideon Smith, Vineeth Kumar, Chris Rogers, Catey Bunce, Julia West, Naveed Younis, Alison Rowlands, Peter Lees, Sandra Lees, Emily Doncaster, Betty Williams, Ticiana Criddle, Stephanie Perrett, Lisa Jones.

**Contributors** DMB (Trial PI) and SH (CI) conceived the trial questions and have been involved in all stages of the study design, and together with AEW (Trial Coordinator) and AW (ISDR Programme Manager) participated in writing the protocol, submission to the funding body and application to the ethics committee. SUH is a senior data manager for the CTRC, and DA is the information systems manager for the CTRC. Both contributed to writing the protocol and devised the electronic case report form (CRF) system and data monitoring systems. MG-F is the lead statistician and coinvestigator, and CC is the trial statistician. IS is a leading UK statistician. All participated in writing the protocol and statistical aspects of the study. MMR is the data warehouse manager. MJ is the lead health economist and a coinvestigator, and CJS is the trial health economist. They designed the heath economic analysis. TM is a senior trials manager and a coinvestigator. LH replaced AEW in the CTRC and has been involved in revising the protocol and preparing the manuscript with DMB and CJS. JK and JC are PPI members and contributed to the design of the risk calculation engine and the trial. All authors read and approved the final manuscript.

**Funding** This paper presents independent research funded by the National Institute for Health Research (NIHR) under the Programme Grants for Applied Research programme (RP-PG- 1210-12016).

**Disclaimer** The views expressed are those of the authors and not necessarily those of the NHS, the NIHR or the Department of Health.

**Competing interests** None declared.

**Patient consent for publication** Not required.

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
