## [Reviewer comments · BMJ Open]

ARTICLE DETAILS

TITLE (PROVISIONAL)	Individualised Screening for Diabetic Retinopathy: the ISDR study. Rationale, design and methodology for a randomised controlled trial comparing annual and personalised variable-interval risk-based screening
AUTHORS	Broadbent, Deborah; Sampson, Christopher; Wang, Amu; Howard, Lola; Williams, Abigail; Howlin, Susan; Appelbe, Duncan; Moitt, Tracy; Cheyne, C; Rahni, Mehrdad; Kelly, John; Collins, John; García-Fiñana, Marta; Stratton, Irene; James, Marilyn; Harding, Simon

VERSION 1 - REVIEW

REVIEWER	Pedro ROMERO AROCA Hospital Universitario Sant Joan Avenida Doctor Josep Laporta 2 43206 Reus, SPAIN
REVIEW RETURNED	02-Sep-2018

GENERAL COMMENTS	Revision of the manuscript entitled: "Individualised Screening for Diabetic Retinopathy: the ISDR study. Rationale, design and methodology for a randomised controlled trial comparing annual and personalised retinal microvasculature variable interval risk-based screening." ID 2018-025788 Declaration of interest: I am no conflicts of interest in the review of the present manuscript A. Summary The authors present a study proposal to develop a protocol for personalized screening of patients with diabetic retinopathy, to detect those patients with DR. B. Strengths: The authors currently have a cohort of 4460 patients that will serve as sample size. Also, the seven centers where the study will have extensive experience in screening for diabetic retinopathy (DR). Authors developed a generalizable risk calculation engine (RCE) to assign personalized intervals to DR screening. C. Commentaries. Weaknesses of this study are:
--

	1. The follow-up time of the study is only 36 months, taking into account that the DR is dependent on the duration of the diabetes, I believe that 3 years of follow-up may be too little, a study proposed at seven years would probably be more effective. 2. Patients with diabetes have diabetic retinopathy throughout his illness, such as chronic complication. The appearance of DR depends on the duration of DM and metabolic control usually measured by HbA1c levels. But despite all between 15% and 20% of patients never develop DR, this is due to a likely bad known genetic factor, which causes any risk calculation engine, fails at least 20% of cases, which may limit its usefulness. 3. Just keep in mind that not all patients with diabetes mellitus will respond in the same way to the disease, as Cunha Vaz et al determined, there are three different phenotypes of patients with DM, those who develop DR slowly and progressive, those predominating ischemia and those in which predominates exudation, this also limits the usefulness of CER, since the three phenotypes depend on poorly understood genetic components. Resume. Despite the limitations I've described on this study, the realization of it is interesting, because the DM is a chronic disease of high prevalence and any effort to streamline the screening of DR in diabetic patients is important.
--	--

REVIEWER	Amber van der Heijden VUMC, the Netherlands
REVIEW RETURNED	03-Oct-2018

GENERAL COMMENTS	This is a clearly written design paper, describing an RCT on personalized screening for retinopathy in people with type 2 diabetes. The topic is highly relevant, since it will add knowledge to the field of sustainable care for type 2 diabetes based on personalized approaches. Some comments: 1) Page 8, 15 Why was a new retinopathy risk score developed? Several prediction models for DR are already available, which have been validated in external cohorts, showing good performance. Maybe it is of interest to mention the performance of this new model. 2) The results of this RCT would be more relevant on an international level when the extension of the individualized screening intervals would be larger than a maximum of 24 months. 3) Page 17 - In the sample size calculation, it might be appropriate to take into account the clustering of participants within the same institutes. 4) Is the economic evaluation performed, using a societal perspective? How are data on costs collected? 5) Page 17 – why would differences in attendance rate occur between the retinopathy groups (one eye affected, two eyes affected) please elaborate on this. 6) In the entire paper, use the term 'people with diabetes' instead of patients. 7) Page 9, line 19, correct sentence. 8) Page 15, line 12, is this sentence correct?
--

VERSION 1 – AUTHOR RESPONSE

Response to reviewers

Thank you for your helpful suggestions. We have answered your queries below and made appropriate modifications to the manuscript. We have also addressed the editorial requests.

Reviewer: 1

Reviewer Name: Pedro ROMERO AROCA

Institution and Country: Hospital Universitario Sant Joan, Reus, SPAIN A. Summary

The authors present a study proposal to develop a protocol for personalized screening of patients with diabetic retinopathy, to detect those patients with DR.

B. Strengths:

The authors currently have a cohort of 4460 patients that will serve as sample size. Also, the seven centers where the study will have extensive experience in screening for diabetic retinopathy (DR).

Authors developed a generalizable risk calculation engine (RCE) to assign personalized intervals to DR screening.

C. Commentaries.

Weaknesses of this study are:

1. The follow-up time of the study is only 36 months, taking into account that the DR is dependent on the duration of the diabetes, I believe that 3 years of follow-up may be too little, a study proposed at seven years would probably be more effective.

Response

We agree that a longer period would have provided additional important information. The trial is part of a 5 year NIHR Programme Grant for Applied Research. The primary outcome was to assess whether the trial was safe in terms of attendance rates i.e. would a longer screen interval adversely affect attendance for follow up screening because patients perceived that it was no longer important. 36 months was chosen to ensure that all patients on 24 month intervals would be recalled during the trial.

There are 7 workstreams in the trial (see summary of grant below). The trial could only commence once the risk calculation engine had been developed, GPs and patients had been contacted to obtain their consent to collect data in the warehouse and data flows (including modifications to the screening software) had been completed. Whilst the trial ends at the 3 year follow up point the intention is to use the trial data alongside the cohort data and the health economic model (based on the risk calculation engine) to project the findings of the study into the future.

A further plan at the end of the trial is to pilot the process in a designated population (such as the Liverpool Diabetic Eye Screening Programme which has a screening population of 23,000 people with diabetes) and to present our evidence to the policy makers.

Summary of Programme of Allied Research (RP-PG-1210-12016) extracted from application

“This five year research programme aims to develop a major enhancement to the existing delivery of screening for sight threatening diabetic retinopathy (DR, STDR) for implementation in the NHS. Important evidence gaps in screening for DR will be tackled comprising a collection of prospective

patient centred data in a whole population cohort study and new knowledge produced on outcomes relevant to patients such as visual impairment (VI). An individual risk-based approach will be developed and evaluated based on evidence collected from a well-established diabetes eye care pathway in Liverpool. Safety, cost-effectiveness and acceptability to patients and staff will be measured. Our mixed quantitative and qualitative population-based approach has been piloted in a NIHR funded programme development grant (PDG) and barriers to success identified and overcome. Inter-linked workstreams (WS) will run in parallel, each informing the progress of others: systematic review; establishment of a study data warehouse; prospective observational study; risk calculation development and testing; RCT standard and test screening interval protocols; health economics modelling; exploring perceptions of screening and variable screening intervals amongst PWD and health care staff.

Outputs. We will deliver a tested individualised screening system based on contemporaneous pragmatic risk data ready for implementation in the NHS providing new whole population data on progression to patient centred outcomes as well as patient perceptions of risk. Results will be applicable to other research questions and services.

2. Patients with diabetes have diabetic retinopathy throughout his illness, such as chronic complication. The appearance of DR depends on the duration of DM and metabolic control usually measured by HbA1c levels. But despite all between 15% and 20% of patients never develop DR, this is due to a likely bad known genetic factor, which causes any risk calculation engine, fails at least 20% of cases, which may limit its usefulness.

Response

We appreciate that some patients with poor control never develop DR and likewise others with excellent control develop progressive changes and agree that this is likely to be due to as yet unidentified genetic factors. However in clinical practice we always consider modifiable risk factors and it therefore seems logical to consider these as predictors. The trial should identify whether our risk predictions are sufficiently accurate. Because the data used to develop the risk calculation engine is taken from the local population it should reflect local risk.

3. Just keep in mind that not all patients with diabetes mellitus will respond in the same way to the disease, as Cunha Vaz et al determined, there are three different phenotypes of patients with DM, those who develop DR slowly and progressive, those predominating ischemia and those in which predominates exudation, this also limits the usefulness of CER, since the three phenotypes depend on poorly understood genetic components.

Response

We are very aware of this research.

Resume.

Despite the limitations I've described on this study, the realization of it is interesting, because the DM is a chronic disease of high prevalence and any effort to streamline the screening of DR in diabetic patients is important.

Reviewer: 2

Reviewer Name: Amber van der Heijden

Institution and Country: VUMC, the Netherlands Please state any competing interests or state 'None declared': None declared

Please leave your comments for the authors below

This is a clearly written design paper, describing an RCT on personalized screening for retinopathy in people with type 2 diabetes. The topic is highly relevant, since it will add knowledge to the field of sustainable care for type 2 diabetes based on personalized approaches.

Some comments:

1) Page 8, 15 Why was a new retinopathy risk score developed? Several prediction models for DR are already available, which have been validated in external cohorts, showing good performance.

Maybe it is of interest to mention the performance of this new model.

Response

Work on the Liverpool risk calculation engine (RCE) began in 2010 and has been presented nationally and internationally. We are aware of other predictive models, in particular work on the model in the Netherlands, and have discussed the relative merits of other models compared to ours in our paper on the risk calculation engine 1 published in Diabetologia in 2017. Briefly, the principle differences between our model and those developed by others is that it uses contemporaneous data from people in the same population as the individual to whom it is applied and is applicable to both type 1 and type 2 diabetes. This makes the Liverpool RCE unique. A modelling approach (i.e. using a risk engine which may give a different model in another population or another model 5 years on) needs to be validated in other populations.

We have added the following sentence describing its performance to the manuscript. "The results suggested that implementing personalised risk-based intervals would reduce the number of screen episodes by 30%."

2) The results of this RCT would be more relevant on an international level when the extension of the individualized screening intervals would be larger than a maximum of 24 months.

Response

Safety is the major concern and for this reason the primary outcome is attendance.

In our original incidence paper 2 we suggested that some patients could safely be screened at up to 5 year intervals. In Sweden, based on our data, a 3 year interval has been introduced. However in the UK serious concerns have been raised over the safety of extended intervals and for this reason we were conservative and opted for 24 months.

3) Page 17 - In the sample size calculation, it might be appropriate to take into account the clustering of participants within the same institutes.

Response

The randomisation was stratified by centre so there is no need for adjustment in the statistical analysis. For the sample size calculation, we did not account for "clustering of participants within the same institutes."

We have revised the statistical analysis as below:

"The primary aim is to assess the relative safety of personalised risk-based interval screening as measured by equivalence in attendance rates. The analysis will test the hypothesis that the difference in attendance rates at the first follow-up between the two pathways is within the acceptable range

$\delta=0.05$. If the bounds of the 90% confidence interval for the difference in attendance rate are found to be within the confidence limits $[-\delta, \delta]$ (i.e. $[-0.05, 0.05]$) the results would support equivalence.

The estimated minimum number of patients required is 3940. With an expected loss to follow-up rate of 6% per year due to death and other exclusion from screening (note that non-attendance is the primary outcome and therefore is not factored here) the target for recruitment (randomised into the trial) was 4460 patients ($4460 \times 0.94 \times 0.94 = 3940$).

A secondary aim is to investigate whether personalised screening can be considered as non-inferior in detection of STDR when compared to annual screening. The STDR detection rate predicted for the usual care pathway during the two years follow-up is approximately 6%, based on data from the Liverpool DESP. The sample size required to address the first question ($n=4460$ patients randomised with 3940 patients retained after 2 years from baseline) will permit us to test for non-inferiority in STDR detection with a maximum allowable reduction of 1.5% of the personalised care pathway compared to standard care, with 5% significance level and power between 60-65%.

We will undertake sub-group analyses to assess differences in attendance rates between the two arms for the three different retinopathy groups (the risk groups will be defined based on the individual baseline estimated risk of developing STDR). A logistic mixed-effects model that takes into account the patient's covariate information over time (including HbA1c, systolic blood pressure, total cholesterol, disease duration, retinopathy level, ethnicity, age, gender, smoking status, diabetes type), and screening clinic (clustered data) will be fitted with attendance (Yes/No) at the first screening visit as the outcome variable. The random effects of the model will account for the variability by screening clinic."

4) Is the economic evaluation performed, using a societal perspective? How are data on costs collected?

Response

The economic evaluation will be performed using an NHS and personal social services perspective as recommended by NICE. We have however collected data on the costs to the patient of attending for a screening appointment, such as travel cost and time off work for themselves and anyone accompanying the visit. Whilst the work cannot scope a full societal cost of diabetic retinopathy it will seek to present a broad perspective on the costs of screening itself. The incremental cost effectiveness ratio (ICER) will be calculated for each of the arms of the trial usual care (12 months and risk based interval screening) The cost of the intervention (i.e. the incremental cost of implementing variable-interval screening) will be estimated based on the relevant costs incurred as part of the research programme, relating to the use of the risk calculation engine in practice. The cost per screening episode was calculated on a sub-sample of 868 patients in the main trial by undertaking a detailed micro costing exercise at each of the screening clinics and administering patient level cost questionnaires to the attendees.

We have revised the economic evaluation as below:

"Using screening activity data stored in the ISDR DW and information collected from the visit questionnaire and EQ-5D-5L and HUI3 responses, we will estimate the cost per QALY within the study period associated with risk-based and annual screening from an NHS perspective and where possible incorporate a broader perspective. We will present bootstrapped incremental cost effectiveness ratios and cost effectiveness acceptability curves to characterise the uncertainty associated with our estimates.

We will apply sensitivity analyses to check the sensitivity of the results on the assumption that missing data are missing at random. Different scenarios for missing data mechanisms will be explored."

5) Page 17 – why would differences in attendance rate occur between the retinopathy groups (one eye affected, two eyes affected) please elaborate on this.

Response

The retinopathy level does not affect attendance; it is used to determine risk. Patients with retinopathy in both eyes are at higher risk of progression than patients with retinopathy in only one eye. This is discussed in detail in the risk calculation engine paper1.

6) In the entire paper, use the term ‘people with diabetes’ instead of patients.

Done, thank you.

7) Page 9, line 19, correct sentence.

This sentence is correct.

8) Page 15, line 12, is this sentence correct?

Typo corrected, thank you.

References

Eleuteri A, Fisher AC, Broadbent DM, García-Fiñana M, Cheyne CP, Wang A, Stratton IM, Gabbay M, Seddon D, Harding SP for the ISDR Study Group. Individualised variable interval risk-based screening for sight threatening diabetic retinopathy – the Liverpool Risk Calculation Engine. Diabetologia 2017, 60(11): 2174-2182

Younis N, Broadbent DM, Vora JP, Harding SP. Incidence of sight-threatening retinopathy in patients with Type 2 diabetes in the Liverpool Diabetic Eye Study: a cohort study. Lancet 2003, 361:195-200

VERSION 2 – REVIEW

REVIEWER	Pedro Romero-Aroca Hospital Universitario Samt Joan , Reus , Spain
REVIEW RETURNED	20-Nov-2018

GENERAL COMMENTS	Revision of the manuscript entitled: “Individualised Screening for Diabetic Retinopathy: the ISDR study. Rationale, design and methodology for a randomised controlled trial comparing annual and personalisedetinal microvasculature variable interval risk-based screening.” ID 2018-025788 R1 Declaration of interest: I am no conflicts of interest in the review of the present manuscript A. Summary The authors present a study proposal to develop a protocol for personalized screening of patients with diabetic retinopathy, to detect those patients with DR.
--

	B. Strengths: Authors developed a risk calculation engine (RCE) to assign personalized intervals to DR screening. C. Commentaries. All my questions have been answered correctly Resume. The present study is interesting and well designed, it can be accepted to publish the new version.
--	---